# ATP Production Relies on Fatty Acid Oxidation Rather than Glycolysis in Pancreatic Ductal Adenocarcinoma

**DOI:** 10.3390/cancers12092477

**Published:** 2020-09-01

**Authors:** Jae-Seon Lee, Su-Jin Oh, Hyun-Jung Choi, Joon Hee Kang, Seon-Hyeong Lee, Ji Sun Ha, Sang Myung Woo, Hyonchol Jang, Ho Lee, Soo-Youl Kim

**Affiliations:** 1Division of Cancer Biology, Research Institute, National Cancer Center, Goyang 10408, Korea; ljs891109@gmail.com (J.-S.L.); sujiniii225@gmail.com (S.-J.O.); labchoihj@gmail.com (H.-J.C.); wnsl2820@gmail.com (J.H.K.); shlee1987@gmail.com (S.-H.L.); jsha9595@gmail.com (J.S.H.); hjang@ncc.re.kr (H.J.); 2Center for Liver and Pancreatobiliary Cancer, National Cancer Center, Goyang 10408, Korea; wsm@ncc.re.kr; 3Graduate School of Cancer Science and Policy, National Cancer Center, Goyang 10408, Korea; ho25lee@ncc.re.kr

**Keywords:** PDAC, glycolysis, fatty acid oxidation, ATP production, KC mouse

## Abstract

**Simple Summary:**

We found a conserved pathway for the energy supply in cancer. The present study revealed that glucose is not a major source of ATP production, whereas fatty acid is a major source of electrons for ATP production through fatty acid oxidation (FAO) and oxidative phosphorylation (OxPhos) in cancer cells. NADH is mainly recruited from FAO, which is used for ATP synthesis via OxPhos with electron transfer complexes in cancer cells while ATP is synthesized from NADH produced in the TCA cycle in the normal cells. Therefore, a calorie-balanced low-fat diet showed 3-fold reduction of tumor formation, whereas a high-fat diet caused a 2-fold increase of tumor to compare to the control in a human pancreatic ductal adenocarcinoma xenograft and a homograft KC models.

**Abstract:**

Glycolysis is known as the main pathway for ATP production in cancer cells. However, in cancer cells, glucose deprivation for 24 h does not reduce ATP levels, whereas it does suppress lactate production. In this study, metabolic pathways were blocked to identify the main pathway of ATP production in pancreatic ductal adenocarcinoma (PDAC). Blocking fatty acid oxidation (FAO) decreased ATP production by 40% in cancer cells with no effect on normal cells. The effects of calorie balanced high- or low-fat diets were tested to determine whether cancer growth is modulated by fatty acids instead of calories. A low-fat diet caused a 70% decrease in pancreatic preneoplastic lesions compared with the control, whereas a high-fat diet caused a two-fold increase in preneoplastic lesions accompanied with increase of ATP production in the Kras (G12D)/Pdx1-cre PDAC model. The present results suggest that ATP production in cancer cells is dependent on FAO rather than on glycolysis, which can be a therapeutic approach by targeting cancer energy metabolism.

## 1. Introduction

Warburg found that cancer cells produce lactate while normal cells produce CO_2_ from glucose. He misunderstood that mitochondrial destruction obligates cancer cells to depend on glycolysis as the major source of ATP production. Due to the mitochondrial destruction idea, active oxidative phosphorylation (OxPhos) was not allowed to explain cancer cell metabolism [1,2,3,4]. However, the mitochondria of cancer cells are intact as well as the respiratory rate of cancer cells is higher than that of normal cells [5,6]. Cancer cells depend on mitochondrial OxPhos for ATP production. Disruption of mitochondrial oxidative OxPhos function by knockdown of mitochondrial transcription factor A decreases tumorigenesis in an oncogenic *Kras*-driven mouse model of lung cancer [7]. Glycolysis produces only two moles of ATP from one mole of glucose in cancer cells, whereas the tricarboxylic acid (TCA) cycle in normal cells can produce 36 moles of ATP. However, the average contribution of glycolysis to ATP production in cancer cells is approximately 17%, which is 1.4% of total ATP production in hepatoma [8]. The mitochondrial membrane potential of cancer cells is similar to that of normal cells, whereas the glucose-driven TCA cycle is stalled [9,10,11,12,13,14]. This suggests that intermediates of the TCA cycle are supplied by other carbon sources in cancer cells. Alternatively, NADH or FADH_2_ is supplied from byproducts such as the dehydrogenase reaction in the cytosol. According to the report on the contribution of pathways to the supply of ATP from other cancer cells, OxPhos is responsible for approximately 90% of ATP production despite an increase in the glycolysis rate [15,16,17]. In OxPhos, the transfer of electrons from NADH or FADH_2_ to O_2_ by a series of electron carriers results in the generation of ATP. OxPhos is active for ATP synthesis in cancer cells, although the major source of NADH and FADH_2_ remains unidentified [18].

We found that cancer cells prefer to use cytosolic NADH as an electron donor for ATP production through electron transfer complex [12,13,14,19,20,21,22]. To block the transfer of cytosolic NADH into mitochondria, we knocked down SLC25A11, the mitochondrial 2-oxoglutarate/malate carrier protein in the malate aspartate shuttle (MAS). This markedly lowered ATP production and inhibited the growth of cancer cells, which was not observed in normal cells. [22]. These findings suggest that MAS is a major contributor to ATP production in cancer. Exciting therapeutic opportunities unveiled by the regulation of fatty acid oxidation (FAO) have been suggested [23]. FAO generates NADH, FADH_2_, and acetyl CoA from the catabolism of a four-carbon fatty acid through cyclic reaction. NADH and FADH_2_ produce ATP by OxPhos through the electron transport chain [23]. An increase of FAO may contribute to cancer cell proliferation through enriched ATP supply because one mole of palmitic acid generates 129 moles of ATP. In this study, we found that ATP levels did not decrease in pancreatic ductal adenocarcinoma (PDAC) cells grown under glucose-free conditions for 24 h. Based on these results, we investigated the major metabolic pathways responsible for ATP production including glycolysis, TCA cycle, MAS, and fatty acid oxidation in pancreatic cancer cells under normal culture condition. In order to inhibit pathways, glucose free media or 2-Deoxy-D-glucose (2-DG) was used for blocking glycolysis [24] and inhibitors were also used as fluoroacetate (FA) for blocking TCA cycle [25], amino-oxyacetic acid (AOA) for blocking MAS [26], and trimetazidine for blocking fatty acid oxidation [27]. All experiments of blocking metabolic pathways were performed under normal culture conditions except glucose free media.

## 2. Results

### 2.1. Glucose Is Not a Source for ATP Production in Cancer Cells

Cancer cells use mainly glucose for ATP production through glycolysis because the TCA cycle in cancer cells is thought to be stalled [1,2,3,4]. To examine whether cancer cells depend on glucose for ATP production, PDAC cell lines were incubated in DMEM containing 10% FBS without glucose for 24 h (Figure 1A). Oxygen consumption rate (OCR) and ATP production increased by 14% and 17%, respectively, in MIA PaCa-2 cells, whereas they decreased in SNU-324 cells by 18% and 22%, respectively (Figure 1A). Similar results were obtained in other cancer cell lines (Appendix A). However, we did not observe change of OCR and ATP production by glucose deprivation in the normal human pancreatic nestin expressing (HPNE) cells (Appendix A). This suggests that normal cells may reroute supply of TCA intermediates from glucose to other nutrients such as amino acid, fatty acid, and glutamine because normal cell operates various metabolic pathways to make a balance of metabolites level. To determine whether cancer cells rely on glycolysis for ATP production, hexokinase was blocked by treatment with 2-deoxyglucose for 24 h, which had no effect on OCR and ATP production in PDAC cell lines (Figure 1B). We observed a decrease of pyruvate and lactate levels by 50% and 84% respectively in MIA PACa-2 cells treated with 2-DG (Appendix A). These results suggest that glucose or glycolysis is not the major source of ATP in cancer cells under normal nutrient conditions. Data indicating that glycolysis is the main pathway for ATP production in cancer are derived from cancer cells grown under limited nutrient conditions, such as in medium containing only glucose [1,2,3,4]. To examine the effect of glucose on the mitochondrial membrane potential in cancer cells, MIA PaCa-2 cells were incubated in media containing 1 mM of 2-deoxyglucose for 48 h, which resulted in a slight decrease of mitochondrial membrane potential less than 10% (Figure 1C). Activation of the mitochondrial membrane potential can be detected by comparing tetramethylrodamine ester (TMRE) staining between cancer cells and human pancreatic nestin-expressing (HPNE) cells under normal culture conditions (normoxia and high glucose) (Appendix A).

### 2.2. Identification of the Metabolic Pathway for ATP Production in PDAC Cells

To identify the main metabolic pathway for ATP production in MIA PaCa-2 cells, four pathways were blocked including glycolysis, TCA, MAS, and fatty acid oxidation (FAO), and changes in metabolites including ATP were measured (Figure 2). Blocking glycolysis by glucose deprivation for 24 h decreased the levels of intermediates of glycolysis by 44% (pyruvate)–90% (3PG) and lactate levels by 80%, whereas ATP levels did not change (Figure 2A). Blocking the TCA cycle by targeting aconitase with fluoroacetate (FA) [25] caused an approximately 20% reduction in ATP production without affecting the metabolic intermediates of the TCA cycle (Figure 2B). We showed that cytosolic NADH levels and MAS, which transports NADH from the cytosol to mitochondria, are increased in tumor cells [22]. Recent study showed that cytosolic NADH production depends greatly on ALDH activity [14,28]. ALDH contributes to productions of fatty acid and NADH using fatty aldehyde formed by lipid peroxidation (LPO) [14]. Later, the cytosolic NADH is transferred into mitochondria through MAS system. Blocking MAS using AOA [29] caused 43% reduction in ATP production and decreased TCA intermediates by 29% (citrate)–62% (malate) (Figure 2C). Therefore, MAS is closely related with FAO, which is considered as a favorable metabolic source for cancer cells. Blocking β-oxidation using trimetazidine, an anti-anginal drug that inhibits 3-ketoacyl-CoA thiolase (acetyl-CoA acylase) [27], reduced ATP production by 48% (Figure 2D). We showed that ALDH3A1 has an important role in lipid catabolism by catalyzing the production of fatty aldehydes by lipid peroxidation in cancer cells [14]. Therefore, an increase of 4-hydroxynonenal by knock down of ALDH3A1 was inversely correlated with ATP production in cancer cells [14]. PDAC cells could produce NADH using ALDHs through lipid peroxidation in the cytosol, and NADH may be transferred into mitochondria via MAS (Figure 2D). This suggests that fatty acids or lipids derived from nutrients under normal culture conditions could constitute a major source of ATP in PDAC.

### 2.3. The TCA Cycle Does Not Affect ATP Production in PDAC Cells

TCA intermediates can be supplied by various sources from nutrients such as amino acids, fatty acids, and carbohydrates. We investigated whether NADH produced by the TCA cycle is responsible for ATP production in PDAC. To inhibit the TCA cycle, we used FA because it disrupts the TCA cycle by forming fluoroacetyl CoA. This compound reacts with citrate synthase to produce fluorocitrate, which binds to and inhibits aconitase, thereby halting the TCA cycle [25]. Blocking the TCA cycle using FA did not reduce OCR and ATP production in PDAC cells (Figure 3A), whereas it caused a 60% and 45% reduction of OCR and ATP production respectively in HPNE cells (Figure 3B). FA treatment had no effect on TCA cycle intermediates such as oxaloacetic acid (OAA) and α-ketoglutarate (α-KG) in MIA PaCa-2 (Figure 3C) whereas OAA and α-KG decreased in HPNE cells by 25% and 28% respectively (Figure 3D), which suggests that ATP production depends on NADH from the TCA cycle in normal cells but not in PDAC. This indicates an important physiological feature of cancer cell metabolism. TCA cycle intermediates may be important for ATP synthesis by mediating NADH production in cancer mitochondria because they can be used for the TCA cycle. However, the present results indicate that PDAC cells do not depend on the TCA cycle to produce NADH for ATP synthesis under normal nutrient conditions.

### 2.4. MAS Is an Important Pathway for ATP Production in PDAC Cells

Lehninger et al. showed that large amounts of cytosolic NADH are oxidized by the mitochondrial respiratory chain in cancer cells via MAS by metabolic flow rate analysis using D-[14C] glucose [29]. Knock down of the oxoglutarate transporter (OGC) greatly reduces ATP production and inhibits the growth of cancer cells, but is not observed in normal cells [22]. The transfer of cytosolic NADH into mitochondria in cancer cells requires the MAS, which is composed of two antiporters, the malate-α-KG antiporter (SLC25A11 or OGC) and the glutamate-aspartate antiporter (SLC25A12 or aspartate-glutamate carrier isoform 1), as well as glutamic-oxaloacetic transaminase 1 and 2 (GOT1 and 2), and malate dehydrogenase 1 and 2 [22,29]. The MAS can be blocked by AOA, which inhibits GOT1 and 2 [30]. In this study, blocking MAS using AOA decreased OCR and ATP production by up to 90% in PDAC cells (Figure 4A), whereas it had no effect on OCR and ATP production in HPNE normal cells (Figure 4B). The HPNE cell line was developed from human pancreatic duct cells by transduction with the hTERT gene [31]. This suggests that ATP production in PDAC cells depends on NADH production mediated by lipid peroxidation in the cytosol [32].

### 2.5. FAO Mediates ATP Production in PDAC Cells

β-oxidation is the catabolic process by which fatty acid molecules are broken down to generate acetyl-CoA and NADH and FADH_2_, which are used in the electron transport chain for ATP production. However, β-oxidation occurs in two locations, the peroxisome in the cytosol and mitochondria using long chain fatty acids and mid-chain fatty acids, respectively. To block both β-oxidation pathways, a common metabolic enzyme of β-oxidation that is active in both mitochondria and peroxisomes is necessary. One such enzyme is 3-ketoacyl CoA thiolase, which catalyzes the thiolysis of 3-ketoacyl CoA between C2 and C3 (β and γ carbons). A drug that targets 3-ketoacyl CoA thiolase is trimetazidine, a fatty acid oxidation inhibitor that functions as an anti-ischemic or anti-anginal metabolic agent and improves myocardial glucose utilization by inhibiting fatty acid metabolism [27]. Blocking β-oxidation using trimetazidine reduced OCR and ATP production by 38% and 40% in MIA PaCa-2 cells, and by 49% and 79% in SNU-324 cells, respectively, and its effect was dose-dependent (Figure 5A). In HPNE cells, trimetazidine decreased OCR and ATP production by approximately 10% (Figure 5B). To determine whether fatty acids such as linoleic acid, oleic acid, and palmitic acid induce ATP production thorough FAO, PDAC cells were treated with supplementary fatty acids and grown under normal culture conditions. All fatty acids added increased OCR and ATP production by up to 49% and 40%, respectively (Appendix A), whereas they had no effect on HPNE cells (Appendix A). The effect of FAO inhibition on the mitochondrial membrane potential in correlation with ATP synthesis was examined by treating MIA PaCa-2 cells with 2.5 mM trimetazidine, which caused an approximately 25% decrease of mitochondrial membrane potential, as determined by analysis of TMRE intensity (Figure 5C). To test whether inhibition of MAS or FAO using AOA or trimetazidine induces cell death, Annexin V staining was analyzed in MIA PaCa-2 and HPNE cells. Treatment of AOA or trimetazidine for 24 h did not induce cell death both in MIA PaCa-2 and normal HPNE cells (Appendix A). Effect of inhibitions of MAS or FAO on cell proliferation was also analyzed by SRB assay under the same condition. Treatment of AOA or trimetazidine for 24 h down regulated proliferation up to 40% only in MIA PaCa-2 cells while the cell proliferation rate was not reduced in normal HPNE cells (Appendix A).

### 2.6. A High-Fat Diet Promotes PDAC Tumor Growth in an In Vivo Model

As shown in Figure 5, FAO is the main pathway for ATP production in PDAC. Therefore, we investigated whether a high-fat diet can promote tumor growth compared with a calorie-balanced diet with carbohydrates and standard levels of dietary fat (Figure 6). Mice were injected with cancer cells and fed different diets for 10 weeks (Figure 6A). The high-fat diet group showed a 2-fold greater rate of tumor growth than control mice fed a standard level of dietary fat (Figure 6B,C) which is concurred with the relative body weight change (Figure 6D). This result was consistent with those of previous studies showing a correlation between a high-fat diet and increased carcinogenesis in various models of cancer, including breast [33] and skin [34] cancers. Although these studies demonstrated that a high-fat diet promotes tumor growth, the metabolic pathway or mechanism underlying the effect of the high-fat diet on cancer cells remains unknown. Indirect effects of a high-fat diet were proposed as possible mechanisms, including decreased immune cell activity, increased prostaglandin synthesis, increased peroxy radicals, and increased membrane fluidity [33]. To test whether fatty acids were used to energy source, NADH and ATP were analyzed in the tumors from mice by targeted LC-MS/MS. Tumors from high-fat diet showed increased levels of NADH and ATP by up to 25% and 34% respectively to compare to the tumors from control diet (Figure 6E). In this study, analysis of metabolites in the plasma of mice fed a high-fat diet for 10 weeks showed higher levels of free fatty acids, including palmitic acid, linoleic acid, oleic acid, and cis-11, 14-eicosatrienoic acid, than those in the control diet group (Figure 6F).

### 2.7. A Low-Fat Diet Decreases PDAC Tumor Growth in a KC Mouse Model

A transgenic mouse model was generated to express physiological levels of oncogenic Kras (Kras^G12D^) in the progenitor cells of the mouse pancreas, which is termed the KC mouse model (Kras^G12D^; Pdx1-cre). KC mice develop preneoplastic lesions that eventually progress to invasive and metastatic pancreatic adenocarcinoma even if the invasive cancer develops at an advanced age (12–15 months) [35] (Figure 7A). Starting at two months of age, KC mice were fed a normal diet, a high-fat diet, or a low-fat diet for 4 months (Figure 7B). The dietary formulations are shown in Appendix A. Preneoplastic lesions were examined histologically by Hematoxylin and Eosin (H&E) staining. Images of H&E-stained pancreatic tissues suggested that a low-fat diet inhibited the progression of acinar ductal metaplasia (ADM) and pancreatic intraepithelial neoplasia (PanIN) by approximately 60% compared with a high-fat diet (Figure 7C). Body weight was 30% higher in mice fed a high-fat diet than in control mice, whereas it was 15% lower in mice fed a low-fat diet than in control mice (Figure 7D). The expression of cytokeratin-19 (CK-19), a ductal epithelial marker, was approximately 75% lower in low-fat diet mice than in high-fat diet mice (Figure 7E). Immunohistochemical staining for alpha-smooth muscle actin (α-SMA) provided evidence of stromal fibrosis, in particular fibroblast density. The α-SMA-positive area was 85% lower in low-fat diet KC mice than in high-fat diet KC mice (Figure 7F). The results of Ki-67 (proliferation marker) staining of the pancreas in KC mice indicated that a high-fat diet increased the proliferation of ADM and PanIN cells compared with a low-fat diet (Figure 7G). Taken together, these results indicate that a high-fat diet promoted the formation of ADM and PanIN lesions in KC mice.

## 3. Discussion

In this study, exposure to glucose-free media did not reduce ATP production or induce cell death, whereas it inhibited lactate production. This result is consistent with previous observations that glucose is required for bio-building block formation [36,37], whereas it is not required for ATP production [38]. A study suggested that cancer cells do not have an absolute need for glucose and can grow in the presence of other carbon sources [39]. In the complete absence of sugar, HeLa cells were grown indefinitely in media containing dialyzed bovine calf serum and supplementary nucleosides [39].

Glycolysis is considered as a major pathway contributing to ATP synthesis in cancer cells. However, in this study, we found that the contribution of glycolysis to ATP production was negligible. Metabolic flux analysis with carbon-13 labeled glucose shows that glucose contributes significantly to ATP synthesis through glycolysis because of a decrease in TCA cycle intermediates. However, this only occurs when cancer cells are incubated with glucose as the sole carbon source without serum [2,40,41]. Different findings may be obtained under physiological conditions in the presence of serum containing amino acids and fatty acids. This culture condition is similar to that used by Dr. Warburg [1], who cultured cells in Ringer solution with only glucose. In this study, cancer cells were grown in enriched media containing 10% serum and high glucose with supplements. Under these conditions, blocking the TCA cycle decreased ATP production only in normal cells, whereas the levels of ATP did not change in cancer cells. Therefore, cancer cells did not rely on the respiratory TCA cycle for energy production, whereas blocking MAS or FAO decreased ATP levels. These results suggest that FAO is a major source of NADH for aerobic phosphorylation that is transported into mitochondria through the MAS system.

To test which metabolic pathway is critical in cancer ATP production, each specific metabolic pathway was blocked by glucose deprivation or inhibitors against glycolysis, TCA cycle, MAS system, and fatty acid oxidation under normal culture condition containing nutrients such as pyruvate, amino acids, and FBS. Cancer cell showed significant decrease of ATP production by inhibition of MAS or fatty acid oxidation (Figure 2, Figure 3, Figure 4 and Figure 5) while normal cell showed significant decrease of ATP production only by inhibition of TCA cycle (Figure 3, Figure 4 and Figure 5). Therefore, fatty acid is a major source of electrons for ATP production, and FAO is a major system for electron supply in cancer cells. This is consistent with a previous report suggesting that OxPhos is the main source of ATP in cancer [8]. Consistently, cutting the currency flow using a calorie-balanced low-fat diet caused an approximately 70% reduction of ADM and PanIN lesions in a KC mouse model.

Reducing the production of glycolytic ATP by knocking out PKM2 (Pyruvate kinase, PKM2) fails to prevent tumorigenesis, which suggests that PKM2 is not required for cancer cell proliferation [42]. Furthermore, most cancer cells rely on mitochondrial OxPhos for the production of ATP instead of glycolysis. Cancer cells support survival by adapting mitochondrial function. In hypoxic cancer cells, mitochondrial OxPhos functions normally when the oxygen concentration is reduced to 0.5% [43]. OxPhos is active in cancer cells and has been proposed as a therapeutic target despite the fact that the TCA cycle activity in cancer cells is stopped [44]. We recently showed that cytosolic NADH levels are higher in cancer cells than in normal cells [12,14,20]. Furthermore, cancer cells use the MAS system to transport cytosolic NADH into mitochondria for OxPhos [18,22].

FAO can occur both in peroxisomes and in mitochondria. A considerable amount of NADH is transferred into mitochondria through the MAS system. Cytosolic fatty aldehyde breakdown catalyzed by ALDH can be a considerable source of NADH, as shown by the effect of ALDH knockdown on decreasing NADH production in cancer cells [12,14,20]. Fatty aldehydes can be produced by fatty acid peroxidation induced by reactive oxygen species in the cytosol [14]. Therefore, cytosolic fatty acids are considered to be a major source of electrons for ATP production. Fatty acids from adipocytes also can be used for ATP production in cancer cells [45]. Fatty acid oxidation (β-oxidation) in mitochondria produces acetyl-CoA for TCA cycle as well as the reducing equivalents NADH and FADH_2_ for ATP production through OxPhos [23]. The results of this study suggest that fatty acids are obtained from nutrients rather than from fatty acid synthesis in cancer cells, as demonstrated by the marked reduction of tumor growth in mice exposed to a fatty acid-limited diet.

We tested the effects of a calorie-balanced high- or low-fat diet in a human xenograft model and in a homograft KC model. The results showed that a low-fat diet significantly suppressed the formation of ADM and PanIN lesions, whereas a high-fat diet caused an approximately 2-fold increase in ADM and PanIN lesions compared with those in the control. The results were consistent with those of previous studies in humans showing that high-fat diets and obesity are strongly associated with a higher incidence and mortality rate of PDAC [46,47,48]. In prospective cohort studies, obesity significantly increased the risk of PDAC [48]. Furthermore, higher body mass index (BMI) values are associated with a higher risk of death from various cancers, including stomach and prostate cancers in men, and breast, uterine, cervical, and ovarian cancers in women [47]. The relationship between BMI and PDAC may be explained in the axis of abnormal glucose intolerance and hyperinsulinemia [48].

In summary, a conserved pathway for the energy supply must exist in cancer cells because all cancer cells preserve to produce lactate from glucose. The results of the present study indicate that glucose is not a major source of ATP production, whereas fatty acid is a major source of electrons for ATP production through FAO and OxPhos.

## 4. Materials and Methods

Detailed methods are provided in the online version of this paper and include the following: reagents, cell culture, immunohistochemical staining, measurement of ATP and metabolites, apoptosis detection, quantification of metabolites by LC-MS/MS, and the preclinical xenograft model, among others.

### 4.1. Mouse Kras Model of Pancreatic Cancer

Pdx1-cre mice and LSL-Kras^G12D^ mice were obtained from the NCI mouse repository (http://mouse.ncifcrf.gov). LSL-Kras^G12D^; Pdx1-cre (KC) mice were generated by crossing LSL-Kras^G12D^ mice with Pdx1-cre mice. This study was reviewed and approved by the Institutional Animal Care and Use Committee of the National Cancer Center Research Institute, which is an Association for the Assessment and Accreditation of Laboratory Animal Care International accredited facility that abides by the Institute of Laboratory Animal Resources guide (protocols: NCC-19-493). The mice were divided into four groups as follows: a control group fed with a normal diet (D12450B, Research Diets, New Brunswick, NJ, USA), a high-fat diet group (D12492, Research Diets, New Brunswick, NJ, USA), a normal diet group (D10001, Research Diets, New Brunswick, NJ, USA), and a low-fat diet group (D00041102, Research Diets, New Brunswick, NJ, USA).

### 4.2. Preclinical Xenograft Tumor Models

Balb/c-nu mice (Orient, Seoul, Korea) aged 6–8 weeks before tumor induction were used. This study was reviewed and approved by the Institutional Animal Care and Use Committee of the National Cancer Center Research Institute (protocols: NCC-19-494). MIA PaCa-2 cells (1 × 10^7^) in 100 μL of PBS were injected subcutaneously into mice using a 1 mL syringe. The mice were divided into two groups, a control group fed with a normal diet (D12450B, Research Diets, New Brunswick, NJ, USA) and a high-fat diet group (D12492, Research Diets, New Brunswick, NJ, USA). Primary tumor size was measured weekly using calipers. Tumor volume was calculated using the following formula: V = (A × B^2^)/2, where V is the volume (mm^3^), A is the long diameter, and B is the short diameter.

### 4.3. Relative Quantification of Metabolites by Liquid Chromatography-Tandem Mass Spectrometry (LC-MS/MS)

Quantification of metabolites was performed by LC-MS/MS with a 1290 HPLC system (Agilent, Santa Clara, CA, USA), Qtrap 5500 (ABSciex, Concord, Ontario, Canada), and reverse phase (Synergi fusion RP 50 × 2 mm) columns (phenomenex, Torrance, CA, USA).

### 4.4. Relative Quantification of Energy Metabolites and Fatty Acyl CoA Using LC-MS/MS

#### Sample Preparation for LC-MS/MS

One million cells were harvested using 1.4 mL cold methanol/H_2_O (80/20, v/v) after sequential washing with PBS and H_2_O. Cells were lysed by vigorous vortexing prior to addition of 100 μL of internal standard (Malonyl-13C3 CoA; 5 μM). Chloroform was added, and metabolites were extracted from the aqueous phase by liquid–liquid extraction. The aqueous phase was dried in a vacuum centrifuge, and the sample was reconstituted with 50 μL of H_2_O/MeOH (50/50 *v/v*) prior to LC-MS/MS analysis.

### 4.5. LC-MS/MS

Metabolites involved in energy metabolism were analyzed by LC-MS/MS with a 1290 HPLC system (Agilent, Santa Clara, CA, USA), Qtrap 5500 (ABSciex, Framingham, MA, USA), and a reverse phase column (Synergi fusion RP 50 × 2 mm). A 3 μL of sample was injected into the LC-MS/MS system and ionized with a turbo spray ionization source. Mobile phases A and B were 5 mM of ammonium acetate in H_2_O and 5 mM of ammonium acetate in methanol, respectively. The separation gradient was as follows: hold at 0% B for 5 min, 0–90% B for 2 min, 90% B for 8 min, 90–0% B for 1 min, and 0% B for 9 min. Liquid chromatography flow was 70 μL/min, except 140 μL/min between 7–15 min, and column temperature was maintained at 23 °C. Multiple reaction monitoring was used in negative ion mode, and the extracted ion chromatogram corresponding to the specific transition for each metabolite was used for quantification. The area under the curve of each extracted ion chromatogram was normalized to that of the extracted ion chromatogram of the internal standard. The ratio of the peak area of each metabolite to that of the internal standard was normalized using the protein content of each sample and was used for relative comparison. Data analysis was performed using Analyst 1.5.2 software (SCIEX, Framingham, MA, USA).

Fatty acyl CoA was analyzed by LC-MS/MS equipped with 1290 HPLC (Agilent, Santa Clara, CA, USA), Qtrap 5500 (ABSciex, Framingham, MA, USA), and reverse phase (Zorbax 300Extend-C18 2.1 × 150 mm) columns (Agilent, Santa Clara, CA, USA). A 3 μL of sample was injected into the LC-MS/MS system and ionized using a turbo spray ionization source. Acetonitrile/H_2_O (10/90) with 15 mM ammonium hydroxide and acetonitrile with 15 mM ammonium hydroxide were used as mobile phase A and B, respectively. The separation gradient was as follows: hold at 0% B for 3 min, 0 to 50% B for 2 min, 50 to 70% B for 5 min, 70 to 0% B for 0.1 min, and 0% B for 4.9 min. LC flow was 200 μL/min, and the column temperature was maintained at 25 °C. Multiple reaction monitoring was used in positive ion mode, and the extracted ion chromatogram corresponding to the specific transition for each fatty acyl CoA was used for quantification. The calibration range for fatty acyl CoA was 0.1–10,000 nM (r^2^ ≥ 0.99). Data analysis was performed using Analyst 1.5.2 software.

### 4.6. Relative Quantification of Metabolites by Liquid Chromatography-Tandem Mass Spectrometry (LC-MS/MS)

#### Sample Preparation for GC-MS

One million cells were harvested in 1 mlcold methanol after sequential washing with PBS. Next, cells were lysed by vigorous vortexing, acidified with HCl (final concentration, 25 mM), and mixed with 50 μL of internal standard (myristic acid-d27; 0.1 mg/mL). The sample was centrifuged at 13,000 rpm for 10 min, and the supernatant was collected in a fresh tube. Next, 3 mL iso-octane was added, and the tube was centrifuged at 4000 rpm for 20 min. Finally, the upper layer was collected and dried under a vacuum.

### 4.7. Fatty Acid Methyl Ester (FAME) Derivatization

The dried sample was reacted with 200 μL of BCl3-MeOH (12% *w/w*; Sigma-Aldrich, St. Louis, MO, USA) at 60 °C for 30 min. Next, 100 μL of H_2_O and 100 μL of hexane were added sequentially and mixed vigorously. The upper phase was collected after resting for 5 min, and 20–30 mg of anhydrous sodium sulfate was added prior to GC/MS analysis. FAMEs (Sigma-Aldrich, St. Louis, MO, USA) were used to generate calibration curves without derivatization.

### 4.8. GC-MS

FAMEs were analyzed in a GC-MS system (Agilent 7890A/5975C, Santa Clara, CA, USA) fitted with a capillary column (HP-5MS; 30 m × 0.25 mm × 0.2 µm). Electron impact ionization was used in positive ion mode, with an injection volume of 1 μL and a split mode ratio of 10:1. Total analysis time was 73.7 min, and the temperature gradient was as follows: hold at 50 °C for 2 min, 50–120 °C at 10 °C/min, 120–250 °C at 3 °C/min, 250 °C for 15 min, 250–300 °C at 35 °C/min, and 300 °C for 5 min. The calibration range was 0.001–10 mg/mL (r2 ≥ 0.99). Data analysis was performed using MSD Chemstation software (Agilent E02.02.1431, Santa Clara, CA, USA).

### 4.9. XF Cell Mito Stress Analysis

Cells were treated with the indicated drug for 24 h. For OCR determination, cells were incubated in XF base medium supplemented with 10 mM of glucose, 1 mM of sodium pyruvate, and 2 mM of L-glutamine, and were equilibrated in a non-CO_2_ incubator for 1 h before starting the assay. The samples were mixed (3 min) and measured (3 min) using the XFe96 extracellular flux analyzer (Seahorse Bioscience, North Billerica, MA, USA). Oligomycin (0.75 µM), FCCP (1 µM), and rotenone/antimycin A (0.5 µM) were injected at the indicated time points. Finally, the OCR was normalized using the SRB assay.

For experiments in Figure 1B, Figure 2B–D, Figure 3, Figure 4 and Figure 5A,B, MIA PaCa-2 cells were grown in high glucose DMEM (SH30243.01; Hyclone, Logan, UT, USA) containing 10% Fetal bovine serum (FBS; SH30919.03, HyClone, Logan, UT, USA) and SNU-324 cells were grown in RPMI 1640 medium (SH30027.01, HyClone, Logan, UT, USA) containing 20% FBS. The normal hTRET-HPNE cells were grown in 75% DMEM without glucose (with additional 2 mM L-glutamine and 1.5 g/L sodium bicarbonate), 25% Medium M3 base containing 5% FBS, 5.5 mM D-glucose, 10 ng/mL human recombinant and 750 ng/mL puromycin.

### 4.10. Immunohistochemistry

Formaldehyde (4%) fixed specimens were paraffin-embedded and cut at a thickness of 4 μm. Sections were dried for 1 h at 56 °C, and immunohistochemical staining was performed with the automated instrument Discovery XT (Ventana Medical Systems, Tucson, Arizona, USA) using the Chromomap DAB Detection kit as follows: sections were deparaffinized and rehydrated with EZ prep (Ventana, Oro Valley, AZ, USA) and washed with reaction buffer (Ventana, Oro Valley, AZ, USA). The antigens were retrieved with heat treatment in pH 6.0 citrate buffer (Ribo CC, Ventana, Oro Valley, AZ, USA) at 90 °C for 30 min for anti-Ki-67 (ab15580; Abcam, Cambridge, UK), CK-19 (ab52625, Abcam, Cambridge, UK), and α-SMA (ab5694, Abcam, Cambridge, UK).

### 4.11. Measurement of Mitochondrial Membrane Potential (∆ψm)

Mitochondrial membrane potential was analyzed by TMRE staining (87917, Sigma-Aldrich, St. Louis, MO, USA). Cells were plated in a 4-well chambered coverglass (155382, Thermo Fisher Scientific, Waltham, MA, USA) in 0.5 mL culture medium. After 24 h, cells were treated with the indicated drug for 48 h at 37 °C. After that, 100 nM of TMRE and 5 μg/mL of Hoechst 33342 were added to the culture medium for 15 min at 37 °C. The 4-well chambered cover glass was placed on the LSM780 Laser Scanning Microscope in the presence of the TMRE and Hoechst 33342 (H1399, Thermo Fisher Scientific, Waltham, MA, USA). Live cell imaging was performed using the LSM780 Laser Scanning Microscope and Axio Observer Z1 (Carl Zeiss, Oberkochen, Germany). The relative intensity of TMRE was normalized to the arithmetic mean intensity (from Zen software 2.6 blue edition).

For experiments in Figure 1C and Figure 5C, high glucose DMEM (SH30243.01; Hyclone, Logan, UT, USA) containing 10% FBS (SH30919.03, HyClone, Logan, UT, USA) was used.

### 4.12. Cell Culture

For the experiments in Appendix A, human cancer cell lines were obtained from American Type Culture Collection (ATCC) and Korean Cell Line Bank. All cells were incubated at 37 °C and maintained in 5% CO_2_. MIA PaCa-2 and Panc-1 were grown in high glucose DMEM (SH30243.01; Hyclone, Logan, UT, USA) containing 10% FBS and penicillin. BxPC-3 and SNU-213 cells were grown in RPMI 1640 medium (SH30027.01, HyClone, Logan, UT, USA) containing 10% FBS and penicillin. SNU-324 cells were grown in RPMI 1640 medium containing 20% FBS and penicillin. Capan-1 cells were grown in IMDM (12440053, Gibco, Logan, UT, USA) containing 20% FBS and penicillin. Capan-2 cells were grown in McCoy’s 5A (16600082, Gibco, Logan, UT, USA) containing 10% FBS and penicillin.

The conditions of culture media in Figure 1, Figure 2A and Appendix A are as follows. For glucose presence experiments, cells were washed with PBS and cultured in high glucose DMEM (11995065, Thermo Fisher Scientific, Waltham, MA, USA) (MIA PaCa-2, Capan-1, Capan-2 and Panc-1) and RPMI 1640 medium (11875093, Thermo Fisher Scientific) (SNU-213, SNU-324, BxPC-3, SW620, MALME-3M, OVCAR5, ACHN, T47-D, H-522, and Huh-7) containing 10% or 20% FBS and penicillin. For glucose starvation experiments, cells were washed with PBS and cultured in glucose free DMEM (11966025, Thermo Fisher Scientific) containing 10% FBS and penicillin and 1 mM of sodium pyruvate. The normal hTERT-HPNE cells were grown in 75% DMEM without glucose (D-5030, Sigma-Aldrich, St. Louis, MO, USA with additional 2 mM L-glutamine and 1.5 g/L sodium bicarbonate), 25% Medium M3 Base (Incell Corp. Texas, USA) containing 5% FBS, 5.5 mM D-glucose (G8270, Sigma-Aldrich, St. Louis, MO, USA), 10 ng/mL human recombinant EGF (E9644, Sigma-Aldrich, St. Louis, MO, USA) and 750 ng/mL of puromycin (P8833, Sigma-Aldrich, St. Louis, MO, USA). For glucose starvation experiments of hTRET-HPNE cells, we did not add 5.5 mM of D-glucose into the media. Genetic alterations of pancreatic cancer cell lines are shown in Appendix A.

### 4.13. FITC Annexin V and Propidium Iodide (PI) Cell Death Detection

Cell death was analyzed using annexin V-FITC apoptosis detection kit (ALX-850-020, Enzo Life Sciences, Farmingdale, NY, USA). Cells were cultured for 24 h in 100 mm dishes and treated with drugs as indicated. Cells were collected, washed with cold PBS, centrifuged at 1500 rpm for 3 min, and resuspended in 1X binding buffer at a concentration of 5 × 10^6^ cells/mL. The solution (100 μL) was transferred (1 × 10^5^) to a 5 mL polystyrene round-bottom tube, and 5 μL of annexin V-FITC and propidium iodide (PI) were added. The cells were gently vortexed and incubated for 15 min at room temperature in the dark. 400 μL of 1× binding buffer was added to each tube, and the samples were analyzed by FACS flow cytometry (BD Falcon, Bedford, MA, USA). Also, Cells were cultured for 24 h in LAB-TEK II 4 well chambered coverglass (Thremo Fisher Scientific, Waltham, MA, USA) and treated with drugs as indicated. Cells were washed with PBS, and incubated in 1× binding buffer. And 5 μL each of annexin V-FITC, PI and 1 μL of hoechst33342 were added. The cells were incubated for 30 min at 37 °C and maintained in 5% CO_2_. The samples were analyzed by LSM780 confocal microscope.

For Appendix A experiments, we used high glucose DMEM containing 10% FBS and penicillin (MIA PaCa-2) and glucose free media containing 75% DMEM without glucose (with additional 2 mM of L-glutamine and 1.5 g/L sodium bicarbonate) and 25% Medium M3 Base containing 5% FBS, 5.5 mM of D-glucose, 10 ng/mL of human recombinant and 750 ng/mL of puromycin.

### 4.14. Sulforhodamine B (SRB) Assay: Cell Proliferation Assay

Cells were inoculated into 96-well microtitre plates in 100 μL of media at plating densities ranging from 5000 to 40,000 cells/well depending on the doubling time of the individual cell line. After cell inoculation, the microtiter plates were incubated for 24 h prior to the addition of the experimental drugs. The drugs were prepared at the appropriate concentrations with 100 μL added to each well. The plates were incubated in CO_2_ incubator. The assay was terminated by the addition of cold TCA. The cells were fixed in situ by gently adding 50 μL of cold 50% (*w/v*) TCA (final concentration: 10% TCA) and incubated for 60 min at 4 °C. The supernatant was discarded, and the plates were washed five times with tap water and then air dried. Sulforhodamine B (SRB) solution (100 μL) at 0.4% (*w/v*) in 1% acetic acid was added to each well, and the plates were then left for 10 min at room temperature. After staining, the unbound dye was removed by washing five times with 1% acetic acid and the plates were air dried. The bound stain was subsequently solubilized with 10 mM of trizma base, and the absorbance was recorded using an automated plate reader at 515 nm.

### 4.15. Measurement of OAA and α-KG

OAA level was monitored using an OAA colorimetric assay kit followed by the manufacturer’s instructions (K659–100, BioVision, Milpitas, CA, USA). α-KG level was monitored using a α-KG colorimetric assay kit followed by the manufacturer’s instructions (K677–100, BioVision Milpitas, CA, USA). The cells (2 × 10^6^) were lysed in 100 μL of assay buffer and centrifuged under ice-cold conditions at 15,000 rpm for 10 min to pellet the insoluble materials. The supernatant was then collected, and 50 μL of this supernatant was added to a 96-well plate. The final volume was topped up to 100 μL/well with assay buffer. Reaction mix was made by provided protocol and the reaction mix was added to each well containing a test sample. Then, the plate was incubated at room temperature for 30 min in the dark, and the OD was measured at 570 nm using a microplate reader.

## 5. Conclusions

Here, we found that FAO is the main metabolic pathway contributing to ATP synthesis in cancer cells (Figure 7). In cancer cells, NADHs are recruited from FAO in the peroxisome in the cytosol and transported into mitochondria through the MAS system, or short/mid-chain fatty acids are transported into mitochondria through the acyl-CoA transporter. The transported NADH and NADH produced by FAO are used for ATP synthesis via OxPhos with electron transfer complexes. In normal cells, ATP is synthesized from NADH produced in the TCA cycle using glucose as a substrate (Figure 8).

In conclusion, all suggested pathways of energy metabolism can sustain cancer cell growth under various conditions. However, a conserved pathway for the supply of energy must exist in cancer. The results of the present study indicate that glucose is not a major source of ATP production, whereas fatty acids are a major source of electrons for ATP production through FAO and OxPhos.

## Figures and Tables

**Figure 1 cancers-12-02477-f001:**
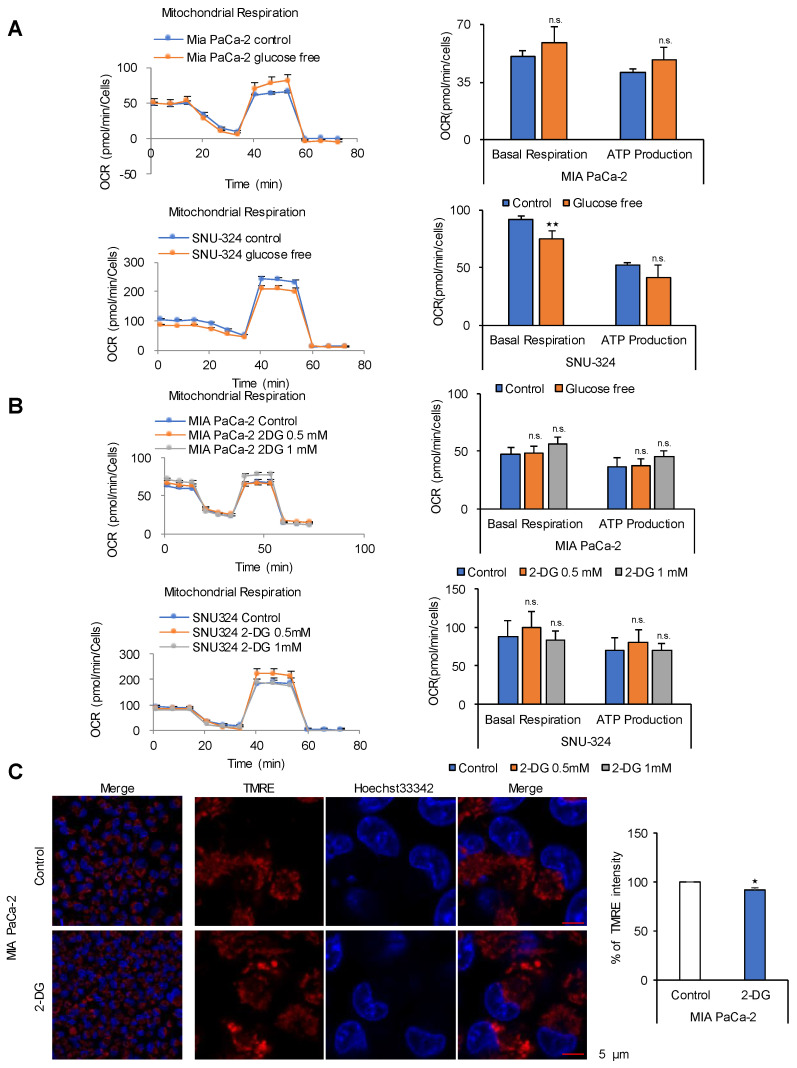
Glucose is not the ATP source in cancer cells. (**A**) Seahorse XF analysis of MIA PaCa-2 and SNU-324 cells treated sequentially with oligomycin, the chemical uncoupler FCCP, and antimycin A/Rotenone. The oxygen consumption rate (OCR) was determined using the Seahorse XFe96 analyzer in normal medium compared with glucose deprivation medium. (**B**) OCR in response to 0.5 mM and 1 mM of 2-DG. Abbreviations: 2-DG, 2-Deoxy-D-glucose. (**C**) MIA PaCa-2 cells were treated with 1 mM of 2-DG for 48 h, and the mitochondrial membrane potential was determined by TMRE staining and live cell imaging. Scale bar = 5 µm. n.s. (not significant), * *p* < 0.05, ** *p* < 0.01 compared with the vehicle control.

**Figure 2 cancers-12-02477-f002:**
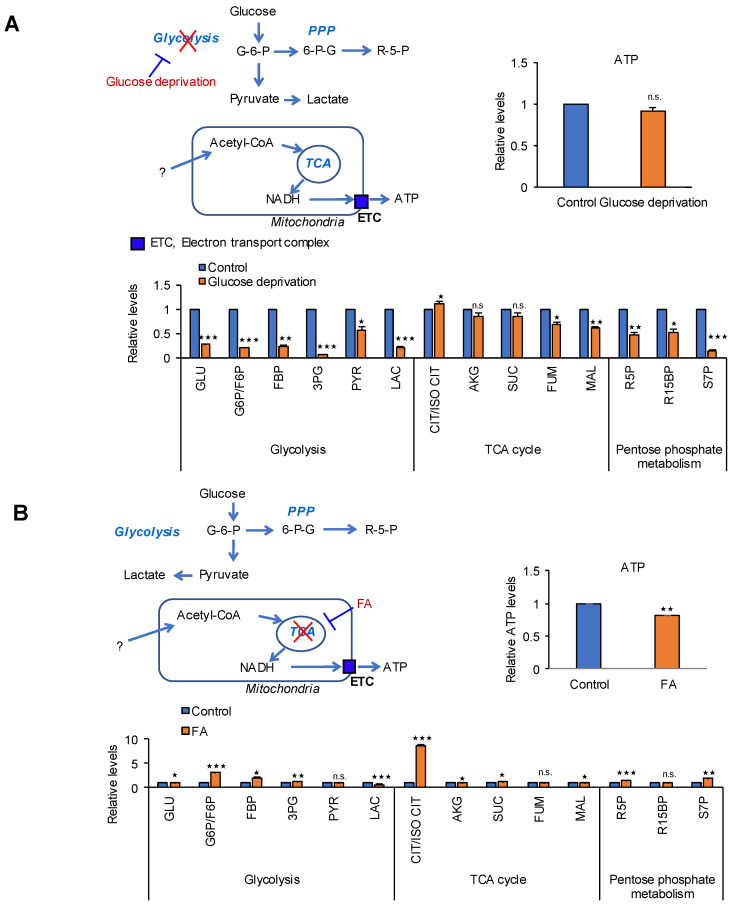
MAS and FAO are major contributors to ATP production in pancreatic cancer cells. Effect of blocking metabolic pathways were analyzed by relative pool sizes of metabolites using targeted LC-MS/MS after various treatments for 24 h. (**A**) glucose deprivation medium for blocking glycolysis, (**B**) 5 mM of fluoroacetate (FA) for blocking TCA cycle, (**C**) 750 µM of amino-oxy acetate (AOA) for blocking MAS system, and (**D**) 2.5 mM of trimetazidine treatments for blocking FAO in MIA PaCa-2 cells. Data represent the mean and standard deviation of three independent experiments. n.s. (not significant), * *p* < 0.05, ** *p* < 0.01, *** *p* < 0.001 compared with the vehicle control.

**Figure 3 cancers-12-02477-f003:**
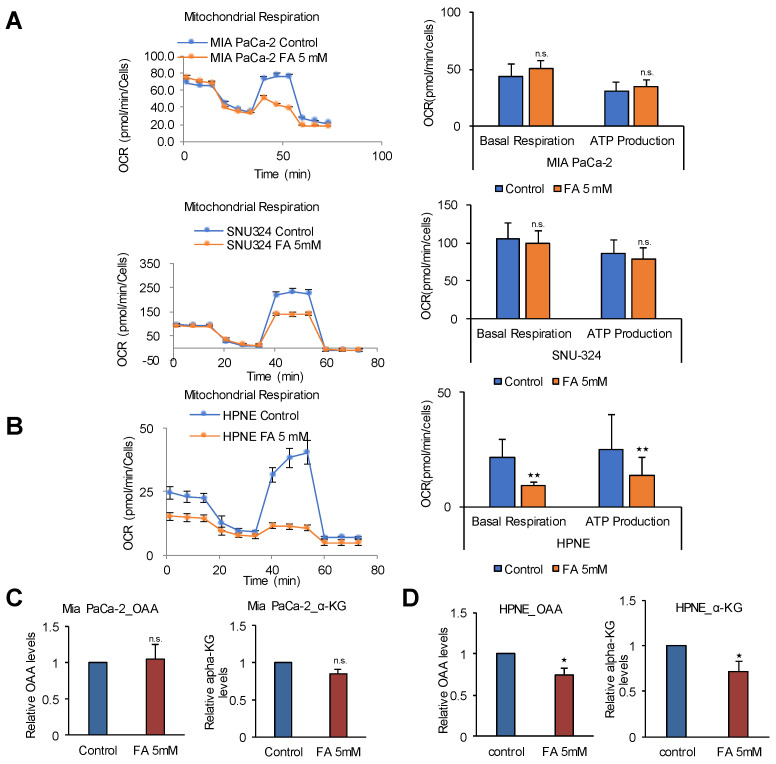
The TCA cycle is not involved in ATP production in PDAC cells. (**A**) OCR was analyzed using the Seahorse XFe96 analyzer. OCR in response to 5 mM of FA in MIA PaCa-2, SNU-324, and (**B**) normal HPNE cells. (**C**) The levels of OAA and α-KG were measured after treatment of MIA PaCa-2 cells with 5 mM of FA for 24 h using OAA colorimetric/fluorometric assay kit and α-KG colorimetric/fluorometric assay kit. (**D**) The levels of OAA and α-KG were measured after treatment of HPNE cells with 5 mM of FA for 24 h. Abbreviations: FA, fluoroacetate. OAA, oxaloacetate. α-KG, α-ketoglutarate. n.s. (not significant), * *p* < 0.05, ** *p* < 0.01 compared with the vehicle control.

**Figure 4 cancers-12-02477-f004:**
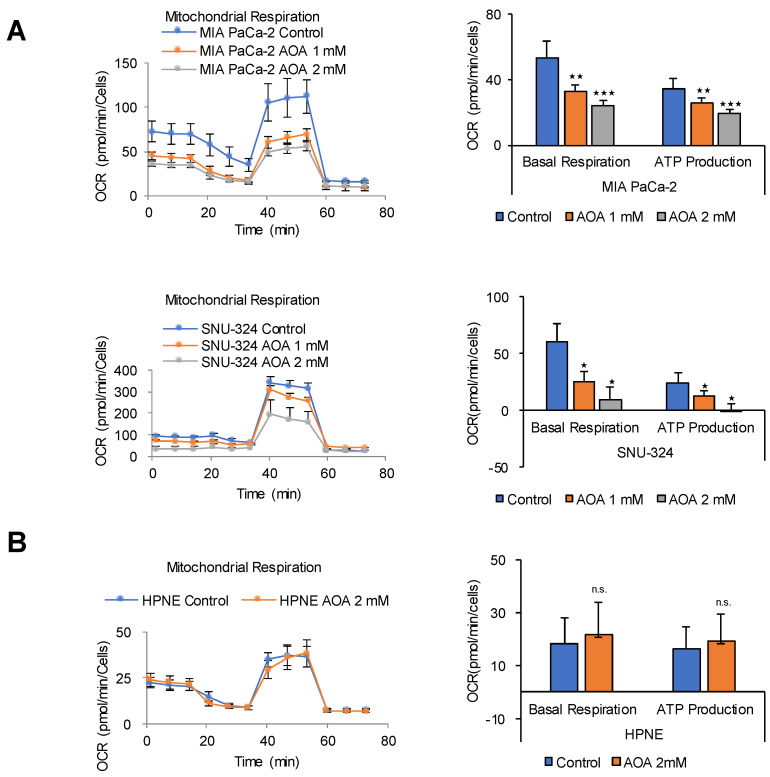
MAS inhibition dose-dependently decreases ATP synthesis in PDAC cells but not in normal cells. (**A**) OCR in response to 1 mM or 2 mM of AOA was analyzed using the Seahorse XFe96 analyzer in MIA PaCa-2, SNU-324, and (**B**) HPNE cells. Abbreviations: AOA, amino-oxy acetate. n.s. (not significant), * *p* < 0.05, ** *p* < 0.01, *** *p* < 0.001 compared with the vehicle control.

**Figure 5 cancers-12-02477-f005:**
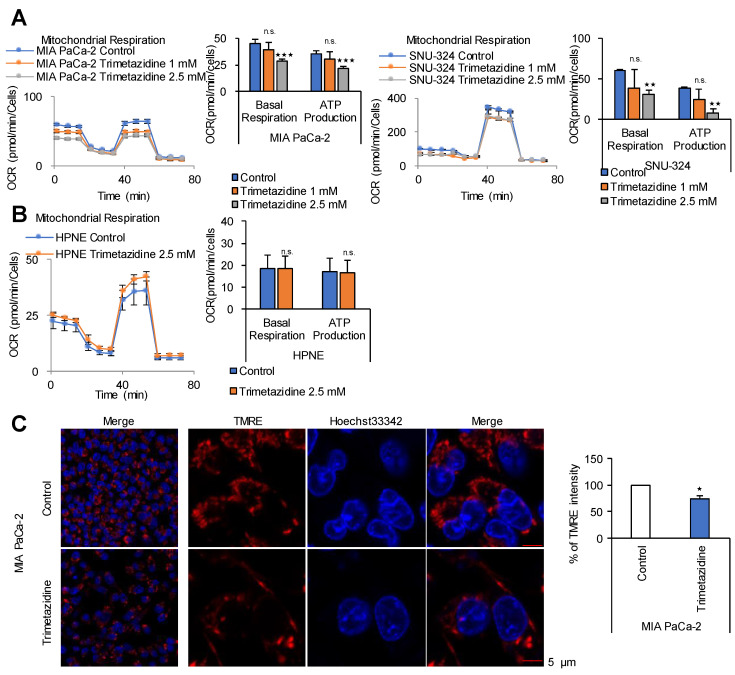
FAO inhibition dose-dependently decreases ATP synthesis in PDAC cells. (**A**) OCR in response to 1 mM or 2.5 mM of trimetazidine was measured using the Seahorse XFe96 analyzer in MIA PaCa-2, SNU-324, and (**B**) HPNE cells. OCR and ATP production in PDAC cells showed dose-dependent reduction while OCR and ATP production in normal cells showed no reduction. (**C**) The mitochondrial membrane potential was determined by TMRE staining and live cell imaging in MIA PaCa-2 cells treated with 2.5 mM of trimetazidine for 48 h. Scale bar = 5 µm. * *p* < 0.05, ** *p* < 0.01, *** *p* < 0.001 compared with the vehicle control. n.s., not significant.

**Figure 6 cancers-12-02477-f006:**
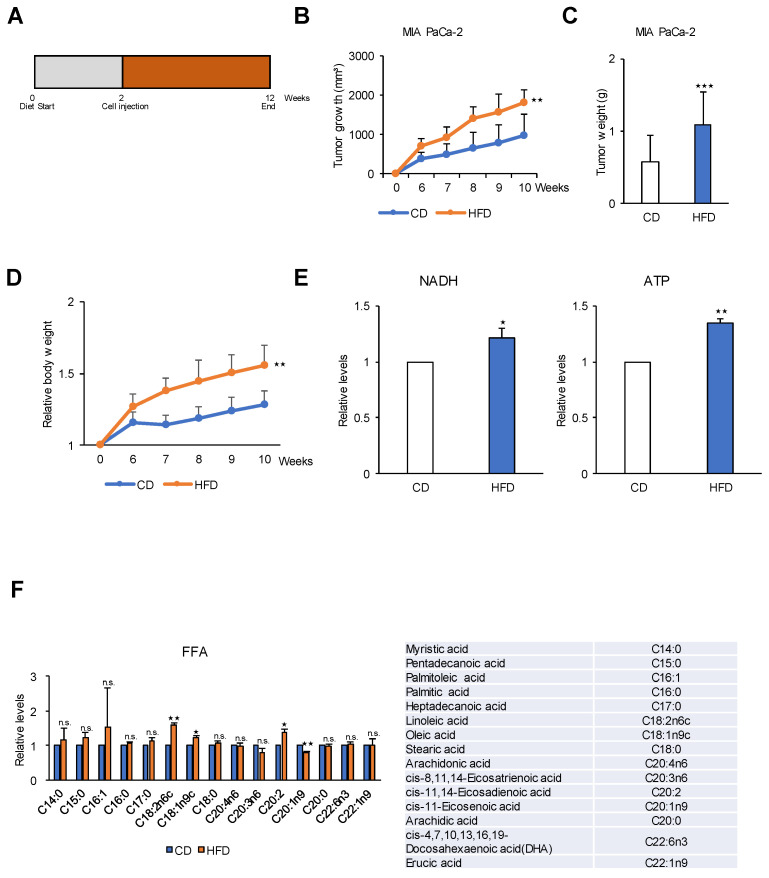
A high-fat diet increases tumor growth in a human pancreatic cancer xenograft mouse model. (**A**) MIA PaCa-2 (1 × 10^7^) cells were injected into 6–8-week-old BALB/c nude mice (*n* = 6 per group) exposed to a HFD or a CD for 12 weeks. (**B**) Tumor growth was measured using calipers. Tumor size is 2-fold greater in HFD group than in the CD group at 10 weeks. (**C**) Final weight of subcutaneous tumors derived from MIA PaCa-2 cells were measured. Tumor weight is 2-fold greater in HFD group than in the CD group at 10 weeks. (**D**) Effect of HFD on body weight in MIA PaCa-2 xenograft mouse model. (**E**) The levels of NADH and ATP were measured in CD and HFD groups by LC-MS/MS analysis. (**F**) Effect of HFD on metabolites in MIA PaCa-2 tumor tissues. Relative pool sizes of metabolites determined by targeted GC-MS. FFA, free-fatty acids; CD, control diet; HFD, high fat diet. * *p* < 0.05, ** *p* < 0.01, *** *p* < 0.001 compared with the control diet. n.s., not significant.

**Figure 7 cancers-12-02477-f007:**
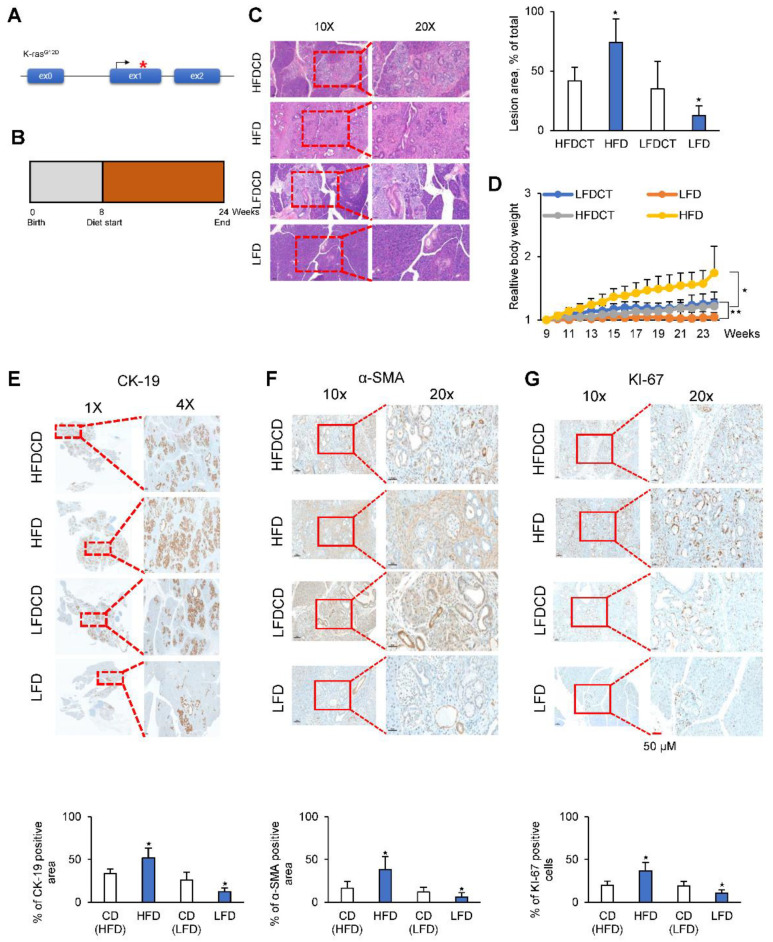
A low-fat diet abrogated spontaneous tumor formation in a KC mouse model. (**A**) Schematic illustration of the genetic construct used to activate *Kras^G12D^* in the pancreas of KC mice. (**B**) Schematic showing the experimental design of diet protocols in KC mice for 24 weeks. (**C**) H&E staining of the pancreas in calorie balanced CD, HFD, and LFD mice, and quantification of the percentage of PanlN lesions (left). Lesion area was analyzed by software and presented as % of the total area (right). (**D**) Changes in body weight of *Kras^G12D^*; *Pdx1*-cre mice for 24 weeks in calorie balanced CD, HFD, and LFD. (**E**–**G**) Staining of the pancreas in CD, HFD, and LFD mice using antibodies of CK-19 as a ductal epithelial marker, α-SMA as a stromal fibrosis marker, and KI-67 as a proliferation marker. LFD, low-fat diet; HFD, high-fat diet; LFDCD and HFDCD, control diets of low fat diet and high fat diet. Scale bar = 50 µm * *p* < 0.05, ** *p* < 0.01.

**Figure 8 cancers-12-02477-f008:**
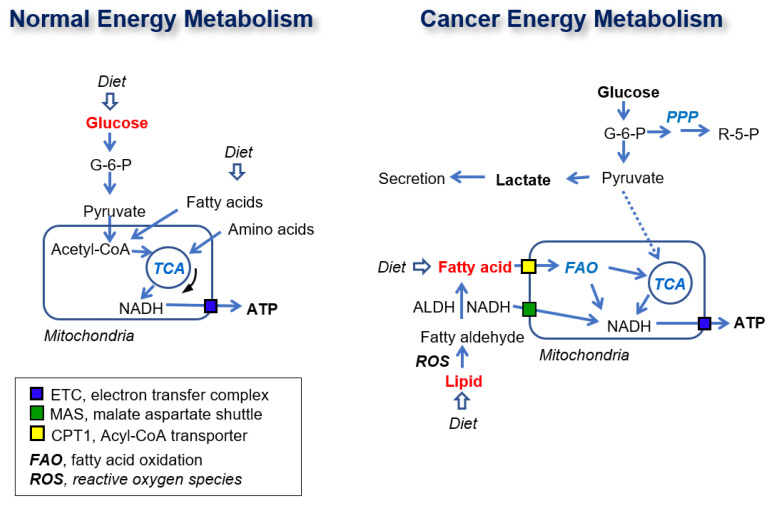
Schematic diagram of normal and cancer energy metabolism. The dotted line presents weak connection. The red color metabolites present major contributor of ATP production.

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
