# Peer review of "ATP Production Relies on Fatty Acid Oxidation Rather than Glycolysis in Pancreatic Ductal Adenocarcinoma"

_cancers, 2020, doi:10.3390/cancers12092477_

Round 1

Reviewer 1 Report

In the manuscript by Jae-Seon Lee et al, the authors found fatty acid oxidation (FAO) serves as a major metabolomics pathway for ATP production in PDAC cell lines rather than glycolysis. Using 2DG to block glycolysis, no effect was observed on OCR and ATP production. It indicates the glucose is not a major source for the ATP production in the PDAC cells. Additional evidence was shown for the minimal role of TCA cycle in the ATP production. The critical role of MAS and FAO was indicated by the more dramatic decrease of ATP production when the PDAC cells were treated with MAS and FAO inhibitors. The correlation between FAO and cell growth was proven by the increased tumor growth in mice with a high-fat diet. In addition, a low-fat diet decreases PDAC tumor growth in a KC mouse model. All these data indicate the correlation between PDAC and fatty acid metabolism and may be due to the ATP production in PDAC cells.

Here are the major concerns:

  1. It is well-known the lipid metabolism is correlated with PDAC progress through activation of kras (Gastroenterology, 2013) or tumor microenvironment modification (Cell, 2020). Actually, the evidences provided in the manuscript are weak for the correlation between lipid metabolism and PDAC via ATP production. To address this point, are there differences for ATP levels in the xenograft tumors with a high-fat diet? Or differences in a genetic mouse model with a low-fat diet? The in vivo parts are not well connected with the in vitro parts.
  2. The authors nicely provide data that MCS and FAO contribute to the major ATP production in PDAC. However, there is no data for the phenotype of cells upon the inhibition of MCS or FAO. How about the cell death when cells treated with MCS or FAO inhibitors? Is there any difference between normal cells and PDAC cancer cells upon treatment?

Here are the minor concerns:

  1. Line 58: What kind of FBS was used here? The glucose concentration is ~5mM in the normal FBS which may be sufficient for cell growth in 24 hours.

  1. Line 63: The concentration of 2DG is too low to block glycolysis at 1mM. The additional assay should be done to validate the block of glycolysis by 2DG such as glucose consumption and lactate production.

  1. Line 85: Remove the space after ATP.

  1. ECAR value should be showed to address if glycolysis is inhibited in the seahorse data in Figure 1.

  1. Is there any difference in the survival of KC mice treated with different diets?

Author Response

Answers to reviewer 1

Here are the major concerns:                                 

  1. It is well-known the lipid metabolism is correlated with PDAC progress through activation of kras (Gastroenterology, 2013) or tumor microenvironment modification (Cell, 2020). Actually, the evidences provided in the manuscript are weak for the correlation between lipid metabolism and PDAC via ATP production. To address this point, are there differences for ATP levels in the xenograft tumors with a high-fat diet? Or differences in a genetic mouse model with a low-fat diet? The in vivo parts are not well connected with the in vitro parts.

Ans) Thank you for your good comments. We have added analysis of ATP production in tumors from high-fat diet at Figure 6E. Tumors from high-fat diet showed increase of NADH and ATP levels by up to 25 % and 34 % respectively to compare to the tumors from control diet (Figure 6E).

  1. The authors nicely provide data that MAS and FAO contribute to the major ATP production in PDAC. However, there is no data for the phenotype of cells upon the inhibition of MAS or FAO. How about the cell death when cells treated with MAS or FAO inhibitors? Is there any difference between normal cells and PDAC cancer cells upon treatment?

Ans) We have added effect of MAS or FAO inhibition on cell death and cell proliferation in MIA PaCa-2 and normal HPNE cells at Supplementary Figure 3. To test whether inhibition of MAS or FAO using AOA or trimetazidine induces cell death, Annexin V staining was analyzed in MIA PaCa-2 and HPNE cells. Treatment of AOA or trimetazidine for 24 h did not induce cell death both in MIA PaCa-2 and normal HPNE cells (Supplementary Figure 3A and B). Effect of inhibitions of MAS or FAO on cell proliferation was also analyzed by SRB assay under the same condition. Treatment of AOA or trimetazidine for 24 h down regulated proliferation up to 40 % only in MIA PaCa-2 cells while cell proliferation was not reduced in normal HPNE cells (Supplementary Figure 3C).

.

Figure S3. Effect of inhibition of malate-aspartate shuttle and fatty acid oxidation on cell proliferation and cell death. (A and B) Cell death analysis using annexin V cell death detection kit was performed with MIA PaCa-2 and normal HPNE cells after inhibitors treatment of aminooxyacetic acid (AOA; malate-aspartate shuttle inhibitor) or trimetazidine (fatty acid oxidation inhibitor) or irinotecan (positive control for cell death) for 24 hrs. (A) Cell death was analyzed by the Flow cytometry and (B) confocal microscope. Cell death was not observed by treatment of AOA or trimetazidine both in MIA PaCa-2 and HPNE while it was observed by treatment of irinotecan. (C) Cell proliferation was measured using SRB assay after MIA PaCa-2 and HPNE cells were treated with AOA or trimetazidine for 24 hrs. Treatment of AOA and trimetazidine down regulated cell proliferation of MIA PaCa-2 but did not reduce the growth of HPNE. Scale bar = 50 µm. * p < 0.05, ** p < 0.01, *** p < 0.001.

Here are the minor concerns:

  1. Line 58: What kind of FBS was used here? The glucose concentration is ~5mM in the normal FBS which may be sufficient for cell growth in 24 hours.

Ans)

We used FBS products from hyclone company (catalog number; SH30919.03). The glucose concentration of FBS is 6.5 mM. When it is added to the medium as 10 %, it is diluted to 0.65 mM, which is not sufficient for cell growth.

  1. Line 63: The concentration of 2DG is too low to block glycolysis at 1mM. The additional assay should be done to validate the block of glycolysis by 2DG such as glucose consumption and lactate production.

Ans) It is a good suggestion to check whether 2DG block the glycolysis.

We added the data of pyruvate and lactate levels in 2-DG treated group at Supplementary Figure 1D. The levels of pyruvate and lactate were measured by targeted LC-MS/MS after MIA PaCa-2 cells were treated with 1 mM of 2-DG for 24 h. We observed decrease of pyruvate and lactate levels by 50 % and 84 % respectively, in MIA PACa-2 cells treated with 2-deoxyglucose (Supplementary Figure 1D). This result suggests that glycolysis was inhibited effectively by 1 mM of 2-DG.

  1. Line 85: Remove the space after ATP.

Ans) Space is removed

  1. ECAR value should be showed to address if glycolysis is inhibited in the seahorse data in Figure 1.

Ans) Answer is the same as the question #4.

We added the data of pyruvate and lactate levels in 2-DG treated group at Supplementary Figure 1D. The levels of pyruvate and lactate were measured by targeted LC-MS/MS after MIA PaCa-2 cells were treated with 1 mM of 2-DG for 24 h. We observed decrease of pyruvate and lactate levels by 50 % and 84 % respectively, in MIA PACa-2 cells treated with 2-deoxyglucose (Supplementary Figure 1D).

  1. Is there any difference in the survival of KC mice treated with different diets?

Ans) It is an interesting question. Here we did not have a chance to test other diets. We will investigate later output difference in the survival of KC mice treated with different diets.

Reviewer 2 Report

This is an interesting paper following a logical protocol by eliminating, in principle, successively different pathway of ATP synthesis in PDAC.

However there are many imprecisions that need to be addressed. For instance the characteristic of the cell lines used should be specified for the reader not in the field. The statistical methods used should be also indicated. The legends of the graphs in Fig.2 are of bad quality. The other figures could be a bit greater in order to be more readable.

In general, the hypotheses should be better argued and discussed by explaining the reasons why other metabolic pathways are not possible.  

Furthermore the interpretation are not always entirely discussed (lipid peroxidation for instance) and perhaps not all hypotheses are envisaged.

The lipid peroxidation could be already discussed and presented in the introduction. A scheme of all NADH producing pathways could be also presented in the introduction.

In Fig.1 the culture medium must be indicated because it is certainly not an only glucose medium. It must be also indicated in the other figures. The effect of glucose free medium in HPNE cells is not shown.

Lines 113-114 and Fig. 3A: On figure 3A in the presence of FA, there is a decrease in respiratory rate after added FCCP which means that the maximal activity of respiratory chain is decreased and lower than the basal respiration. That seems not possible.

In figure 3C what means Panc-1?

Fig. 3, 4 and 5: Is it possible to have negative respiration ? Is there a Seahorse calibration problem? If so, what credit should be given to these figures?

Lines 115-120. I do not very well understand. Is Panc-1 a normal cell line? What is the level of OAA and AKG in normal cells? NADH can be produced from succinate or AKG. This hypothesis should be better explained and argued.

Line 142: Why from lipid peroxidation; because glycolysis is eliminated from Fig.1 ? But is there no other way of NADH production in cancer cells? This hypothesis should be better explained and argued.

Line 157: I guess it is "beta and gamma carbons" and not "beta and beta carbons"

Fig.8 right: if TCA cycle is not used in cancer cells, what is the fate of the acetyl-CoA? Do you see an accumulation of acetyl-CoA in GC-MS?

Fig.8 left: MAS is also operating in normal Energy Metabolism with a similar rate as pyruvate uptake in mitochondria at least for pyruvate produced in glycolysis. Indeed, NADH produced by glycolysis should enter in mitochondria if pyruvate is not converted in lactate. This should be added in figure 8.

If MAS have to be used in normal Energy metabolisme (to regenerate NAD+ in cytosol) because pyruvate enter in mitochondria, that means another assumption/explanation has to be given for figure 4B. Perhaps there is another substrate than glucose in the medium ? This substrate  should not produce NADH in cytosol since inhibtion of MAS does not reduce ATP production and respiration.

Author Response

Answers to reviewer 2

  1. However there are many imprecisions that need to be addressed. For instance the characteristic of the cell lines used should be specified for the reader not in the field. The statistical methods used should be also indicated. The legends of the graphs in Fig.2 are of bad quality. The other figures could be a bit greater in order to be more readable.

Ans)

Thank you for concerns about precisions. We added the information of genetic alterations of pancreatic cancer cell lines at Supplementary Table 2. However, in this study, we did not focus the relationship between gene alteration and metabolism. Rather we have asked a question which metabolism could be a major source of energy in PDAC. Precision study about relationship between genetic alterations and metabolic reliance will be an interesting investigation as a next study.

We have revised Figure 2 with readable bigger font size and rearrangement.

  1. In general, the hypotheses should be better argued and discussed by explaining the reasons why other metabolic pathways are not possible. Furthermore the interpretation are not always entirely discussed (lipid peroxidation for instance) and perhaps not all hypotheses are envisaged. The lipid peroxidation could be already discussed and presented in the introduction. A scheme of all NADH producing pathways could be also presented in the introduction.

Ans) Thank you for your comments.

We have added followings at the introduction,

“These findings suggest that MAS is a major contributor to ATP production in cancer. Unveiled new and exciting therapeutic opportunities by regulation of fatty acid oxidation (FAO) has been suggested [23]. FAO generates NADH, FADH2 and acetyl CoA from the catabolism of a four‑carbon fatty acid through cyclic reaction. NADH and FADH2 produce ATP by OxPhos through the electron transport chain [23]. Increase of FAO may contribute to cancer cell proliferation through enriched ATP supply because 1 mole of palmitic acid generates 129 moles of ATP. In this study, we found that ATP levels did not decrease in pancreatic ductal adenocarcinoma (PDAC) cells grown under glucose-free conditions for 24 h. Based on these results, we investigated the major metabolic pathways responsible for ATP production including glycolysis, TCA cycle, MAS, and fatty acid oxidation in pancreatic cancer cells. In order to inhibit pathways, glucose free media or 2-Deoxy-D-glucose (2-DG) was used for blocking glycolysis [24] and inhibitors were also used as fluoroacetate (FA) for blocking TCA cycle [25], amino-oxyacetic acid (AOA) for blocking MAS [26], and trimetazidine for blocking fatty acid oxidation [27]. All experiments of blocking metabolic pathways were performed under normal culture condition except glucose free media.”

We have added followings at the result section,

“We showed that cytosolic NADH levels and MAS, which transports NADH from the cytosol to mitochondria, are increased in tumor cells [22]. Recent study showed that cytosolic NADH production depends greatly on ALDH activity [14,28]. ALDH contributes to productions of fatty acid and NADH using fatty aldehyde formed by lipid peroxidation (LPO) [14]. Later, the cytosolic NADH is transferred into mitochondria through MAS system. Blocking MAS using AOA [29] caused 43 % reduction in ATP production and decreased TCA intermediates by 29 % (citrate) – 62 % (malate) (Figure 2C). Therefore, MAS is closely related with FAO, which is considered as a favorable metabolic source for cancer cells. Blocking β-oxidation using trimetazidine, an anti-anginal drug that inhibits 3-ketoacyl-CoA thiolase (acetyl-CoA acylase) [27], reduced ATP production by 48 % (Figure 2D). We showed that ALDH3A1 has an important role in lipid catabolism by catalyzing the production of fatty aldehydes by lipid peroxidation in cancer cells [14]. Therefore, increase of 4-hydroxynonenal by knock down of ALDH3A1 was inversely correlated with ATP production in cancer cells [14]. PDAC cells could produce NADH using ALDHs through lipid peroxidation in the cytosol, and NADH may be transferred into mitochondria via MAS (Figure 2D). This suggests that fatty acids or lipids derived from nutrients under normal culture conditions could constitute a major source of ATP in PDAC.”

And discussion session as well.

“To test which metabolic pathway is critical in cancer ATP production, each specific metabolic pathway was blocked by glucose deprivation or inhibitors against glycolysis, TCA cycle, MAS system, and fatty acid oxidation under normal culture condition containing nutrients such as pyruvate, amino acids, and FBS. Cancer cell showed significant decrease of ATP production by inhibitions of MAS and fatty acid oxidation (Figures 2-5) while normal cell showed significant decrease of ATP production only by inhibition of TCA cycle (Figures 3-5).”

  1. In Fig.1 the culture medium must be indicated because it is certainly not an only glucose medium. It must be also indicated in the other figures. The effect of glucose free medium in HPNE cells is not shown.

Ans) Thank you for your important comments. We have expanded method section of cell culture with detail media condition.

4.9. XF Cell Mito Stress Analysis

For experiments in Figure 1B, Figure 2B-D, Figure3, Figure4 and Figure 5A-B, MIA PaCa-2 cells were grown in high glucose DMEM (SH30243.01; Hyclone) containing 10 % fetal bovine serum (FBS; SH30919.03, HyClone) and SNU-324 cells were grown in RPMI 1640 medium (SH30027.01, HyClone) containing 20 % fetal bovine serum. The normal hTRET-HPNE cells were grown in 75 % DMEM without glucose (with additional 2 mM L-glutamine and 1.5 g/L sodium bicarbonate), 25 % Medium M3 base containing 5 % fetal bovine serum, 5.5 mM D-glucose, 10 ng/ml human recombinant and 750 ng/ml puromycin.

4.11. Measurement of mitochondrial membrane potential (∆ψm)

For experiments in Figure 1C and Figure 5C, high glucose DMEM (SH30243.01; Hyclone) containing 10 % fetal bovine serum (FBS; SH30919.03, HyClone) was used.

4.12. Cell culture

For the experiments in supplementary Figure 1, human cancer cell lines were obtained from American Type Culture Collection (ATCC) and Korean Cell Line Bank. All cells were incubated at 37 °C and maintained in 5 % CO2. MIA PaCa-2 and Panc-1 were grown in high glucose DMEM (SH30243.01; Hyclone, Logan, UT, USA) containing 10 % fetal bovine serum and penicillin. BxPC-3 and SNU-213 cells were grown in RPMI 1640 medium (SH30027.01, HyClone, Logan, UT, USA) containing 10 % fetal bovine serum and penicillin. SNU-324 cells were grown in RPMI 1640 medium containing 20 % fetal bovine serum and penicillin. Capan-1 cells were grown in IMDM (12440053, Gibco, Logan, UT, USA) containing 20 % fetal bovine serum and penicillin. Capan-2 cells were grown in McCoy’s 5A (16600082, Gibco, Logan, UT, USA) containing 10 % fetal bovine serum and penicillin.

The conditions of culture media in Figure 1, Figure 2A and Supplementary Figure 1 are as follows; for glucose presence experiments, cells were washed with PBS and cultured in high glucose DMEM (11995065, Thermo Fisher Scientific, Waltham, MA, USA) (MIA PaCa-2, Capan-1, Capan-2 and Panc-1) and RPMI 1640 medium (11875093, Thermo Fisher Scientific) (SNU-213, SNU-324, BxPC-3, SW620, MALME-3M, OVCAR5, ACHN, T47-D, H-522 and Huh-7) containing 10 % or 20 % fetal bovine serum and penicillin. For glucose starvation experiments, cells were washed with PBS and cultured in glucose free DMEM (11966025, Thermo Fisher Scientific) containing 10 % fetal bovine serum and penicillin and 1 mM sodium pyruvate. The normal hTERT-HPNE cells were grown in 75 % DMEM without glucose (D-5030, Sigma-Aldrich, St. Louis, MO, USA with additional 2 mM L-glutamine and 1.5 g/L sodium bicarbonate), 25 % Medium M3 Base (Incell Corp. Texas, USA) containing 5 % fetal bovine serum, 5.5 mM D-glucose (G8270, Sigma-Aldrich, St. Louis, MO, USA), 10 ng/ml human recombinant EGF (E9644, Sigma-Aldrich, St. Louis, MO, USA) and 750 ng/ml puromycin (P8833, Sigma-Aldrich, St. Louis, MO, USA). For glucose starvation experiments of hTRET-HPNE cells, we did not add 5.5 mM D-glucose into the media. Genetic alterations of pancreatic cancer cell lines are shown in Supplementary Table 2.

4.13. FITC annexin V and propidium iodide (PI) cell death detection

Cell death was analyzed using annexin V-FITC apoptosis detection kit (ALX-850-020, Enzo Life Sciences, Farmingdale, NY, USA). Cells were cultured for 24 h in 100 mm dishes and treated with drugs as indicated. Cells were collected, washed with cold PBS, centrifuged at 1,500 rpm for 3 min, and resuspended in 1X binding buffer at a concentration of 5 × 106 cells/ml. The solution (100 μl) was transferred (1 × 105) to a 5 ml polystyrene round-bottom tube, and 5 μl of annexin V-FITC and propidium iodide (PI) were added. The cells were gently vortexed and incubated for 15 min at room temperature in the dark. 400 μl of 1× binding buffer was added to each tube, and the samples were analyzed by FACS flow cytometry (BD Falcon, Bedford, MA, USA). Also, Cells were cultured for 24 h in LAB-TEK II 4 well chambered coverglass (Thremo Fisher Scientific, Waltham, MA, USA) and treated with drugs as indicated. Cells were washed with PBS, and incubated in 1X binding buffer. And 5 μl each of annexin V-FITC, PI and 1 μl of hoechst33342 were added. The cells were incubated for 30 min at 37 °C and maintained in 5 % CO2. The samples were analyzed by LSM780 confocal microscope. For Supplementary Figure 3 experiments, we used high glucose DMEM containing 10 % fetal bovine serum and penicillin (MIA PaCa-2) and glucose free media containing 75 % DMEM without glucose (with additional 2 mM L-glutamine and 1.5 g/L sodium bicarbonate) and 25 % Medium M3 Base containing 5 % fetal bovine serum, 5.5 mM D-glucose, 10 ng/ml human recombinant and 750 ng/ml puromycin.

4.14. Sulforhodamine B (SRB) assay: cell proliferation assay

Cells (100 μl) were inoculated into 96-well microtitre plates at plating densities ranging from 5000 to 40 000 cells/well depending on the doubling time of the individual cell line. After cell inoculation, the microtiter plates were incubated for 24 h prior to the addition of the experimental drugs. The drugs were prepared at the appropriate concentrations in 100 μl was added to each well; the plates were incubated in CO2 incubator. The assay was terminated by the addition of cold TCA. The cells were fixed in situ by gently adding 50 μl of cold 50 % (w/v) TCA (final concentration, 10 % TCA) and incubated for 60 min at 4 °C. The supernatant was discarded, and the plates were washed five times with tap water and then air dried. Sulforhodamine B solution (100 μl) at 0.4 % (w/v) in 1 % acetic acid was added to each well, and the plates were then left for 10 min at room temperature. After staining, the unbound dye was removed by washing five times with 1 % acetic acid; the plates were air dried. The bound stain was subsequently solubilized with 10 mM trizma base, and the absorbance was recorded using an automated plate reader at 515 nm.

We added the data of glucose free medium in HPNE cells at Supplementary Figure 1C and D. However, we did not observe change of OCR and ATP production by glucose deprivation in human pancreatic nestin expressing (HPNE) cells (Supplementary Figure 1C). This suggests that normal cells may reroute supply of TCA intermediates from glucose to other nutrients such as amino acid, fatty acid, and glutamine because normal cell operates various metabolic pathways to make a balance of metabolites level. We observed decrease of pyruvate and lactate levels by 50 % and 84 % respectively, in MIA PACa-2 cells treated with 2-deoxyglucose (Supplementary Figure 1D).

  1. Lines 113-114 and Fig. 3A: On figure 3A in the presence of FA, there is a decrease in respiratory rate after added FCCP which means that the maximal activity of respiratory chain is decreased and lower than the basal respiration. That seems not possible.

Ans)

We repeated Figure 3A experiment. And we observed the same results as before. We found a paper of the same result as ours. (Ref; doi:10.1073/pnas. Nanotechnology-mediated crossing of two impermeable membranes to modulate the stars of the neurovascular unit for neuroprotection, Figure 6B). Treatment of astrocytes (normal cell) with 5-Fluorocitrate (FC) resulted in a significant decrease of basal OCR, basal respiration, and ATP production (Fig. 6B). Also, there is a decrease in respiratory rate after added FCCP which means that the maximal activity of respiratory chain is decreased and lower than the basal respiration. (5-Fluorocitrate- aconitase inhibitor, fluoroacetic acid- aconitase inhibitor)

  1. In figure 3C what means Panc-1?

Ans) we have deleted the Panc-1 because that Panc-1 line was not used throughout this study.

  1. Fig. 3, 4 and 5: Is it possible to have negative respiration? Is there a Seahorse calibration problem? If so, what credit should be given to these figures?

Ans) Thank you for your good comments. We have repeated the experiments with increased number of cells, which resulted in positive respiration. We have replaced Figures 3B, 4A and 5B with new data.

Figure 3B

Figure 4A

Figure 5B

  1. Lines 115-120. I do not very well understand. Is Panc-1 a normal cell line? What is the level of OAA and AKG in normal cells? NADH can be produced from succinate or AKG. This hypothesis should be better explained and argued.

Ans) We have deleted the Panc-1 data because Panc-1 cell line was not used throughout this study.

We added the data of oxaloacetate and alpha-ketoglutarate levels in FA treated group at Figure 3D. The levels of oxaloacetate and alpha-ketoglutarate were measured by oxaloacetate and alpha-ketoglutarate colorimetric/fluorometric assay kit after HPNE cells were treated with 5 mM of FA for 24 h. We observed decrease of oxaloacetate and alpha-ketoglutarate levels by 25 % and 28 % respectively, in HPNE cells treated with FA (Figure 3D).

  1. Line 142: Why from lipid peroxidation; because glycolysis is eliminated from Fig.1 ? But is there no other way of NADH production in cancer cells? This hypothesis should be better explained and argued.

Ans) Thank you for your good comments. The answer is overlapped with question #2.

As we showed about 80 % reduction of lactate by glucose deprivation or 2-DG treatment to compare to the control in Figure 2A or Supplementary Figure 1D, most glucose appears to be used to produce lactate instead of acetyl-coA production. In this study, we have focused on 4 major metabolic pathways including glycolysis, TCA cycle, MAS, and FAO system.

  1. Line 157: I guess it is "beta and gamma carbons" and not "beta and beta carbons"

Ans) It is fixed to g.

  1. Fig.8 right: if TCA cycle is not used in cancer cells, what is the fate of the acetyl-CoA? Do you see an accumulation of acetyl-CoA in GC-MS?

Ans) It is a good question. We have not measured the acetyl-coA. Therefore, we have deleted acetyl-CoA in Figure 8. The fate of Acetyl-CoA will be interesting investigation in next study. However fatty acid oxidation is important pathway supporting cancer ATP production as we showed in this study.

  1. Fig.8 left: MAS is also operating in normal Energy Metabolism with a similar rate as pyruvate uptake in mitochondria at least for pyruvate produced in glycolysis. Indeed, NADH produced by glycolysis should enter in mitochondria if pyruvate is not converted in lactate. This should be added in figure 8.

Ans) The pyruvate entry to TCA was added as a dotted arrow in Figure 8.

  1. If MAS have to be used in normal Energy metabolisme (to regenerate NAD+ in cytosol) because pyruvate enter in mitochondria, that means another assumption/explanation has to be given for figure 4B. Perhaps there is another substrate than glucose in the medium? This substrate should not produce NADH in cytosol since inhibtion of MAS does not reduce ATP production and respiration.

Ans)

Normal cell does not use MAS system for energy production. Cancer cells rely on cytosolic NADH transported through the MAS system for OxPhos, whereas normal cells consume mitochondrial NADH produced by the TCA cycle for OxPhos (EBioMedicine 40 (2019) 184–197). Therefore, MAS knock down did not change any level of ATP production in normal cells.

It was observed that MAS inhibitor treatment in normal cell did not change ATP production while it decreases ATP production in a dose dependent manner in cancer cells in Figure 4B.

Fig (EBioMedicine 40 (2019) 184–197). Cell proliferation assay using SRB assay and ATP assay were performed with IMR90 (normal) cells by transfection of MAS (SLC25A11) siRNA or control siRNA for 48 h. This did not change any level of normal cells.

Round 2

Reviewer 1 Report

Two simple questions:

  1. Please check the seahorse data for the cells treated with 2DG. Is the basal ECAR value changed between the 2DG treated cells and the control cell?
  2. Are there differences in the survival of the mice treated with high-fat, low-fat, and control diet?

Author Response

Reviewer 1

1) Please check the seahorse data for the cells treated with 2DG. Is the basal ECAR value changed between the 2DG treated cells and the control cell?

Ans)

The levels of ECAR (Glycolytic Reserve Value) were measured by XFe96 extracellular flux analyzer after MIA PaCa-2 and SNU-324 cells were treated with 1 mM of 2-DG for 24 h. We observed decrease of ECAR (Glycolytic Reserve Value) levels by 33 % and 80 % respectively, in MIA PACa-2 cells and SNU-324 treated with 2-deoxyglucose 1 mM. This result suggests that glycolysis was inhibited effectively by 1 mM of 2-DG.

2) Are there differences in the survival of the mice treated with high-fat, low-fat, and control diet?

Ans) It is a good question. We used PDX1-Cre, LSL-KrasG12D mouse model. A subset of these mice developed pancreatic ductal adenocarcinoma, with a median overall survival of 14 months (ref; doi:10.1097/PPO.0b013e31827ab4c4.). We finished the diet experiments on 24 weeks (6 months). Therefore, we did not have a chance to look up the change in survival rates.

Reviewer 2 Report

This new version is greatly improved. Particularly the Fig.2 which describes part of the basic experiments done in this paper and the cell culture conditions. This clearly brings up another question using the inhibitors AOA and Trimetazidine: AOA inhibiting the transfer of NADH equivalents in mitochondria should increase NADH /NAD ratio in cytosol and thus decrease the activity of lipid peroxidation and thus FA production in cytosol and thus should also reduce the beta oxidation of FA in mitochondria.

In the same way trimetazidine inhibiting beta oxidation of FA in mitochondria should slow down the incorporation of FA in mitochondria leading to accumulation of FA in cytosol and thus decrease lipid peroxidation in cytosol.

Thus the effects on ATP synthesis of both inhibitors, trimetazidine and AOA are not independent on betaoxidation of FA and on lipid peroxidation respectively. This appears in the similar inhibition of ATP level by each of the inhibitors, 43% and 48%. Otherwise the inhibitions should be additive.

This point is discussed by the authors in the results section:” Therefore, MAS is closely related with FAO, which is considered as a favorable metabolic source for cancer cells.” But the reason for this relationship should perhaps be detailed in the discussion (production of both NADH and FA in the cytosol by lipid peroxidation leading both to NADH and FA oxidation followed by ATP synthesis in mitochondria).

Line 161 : Lehninger instead of Leninger

Line 279, Fig. 8. Perhaps MAS not in red, to be not confused with metabolites with major contribution to ATP production?

Fig 8 right: I would make a combination of the two figures 8 with Fatty Acid -> FAO -> Acetyl-CoA (in TCA cycle)

I would keep the dotted blue line from Pyruvate to Acetyl-CoA and the blue line from (blue FAO) to NADH. A blue line from TCA cycle to NADH could be added.

Author Response

Reviewer 2

This point is discussed by the authors in the results section:” Therefore, MAS is closely related with FAO, which is considered as a favorable metabolic source for cancer cells.” But the reason for this relationship should perhaps be detailed in the discussion (production of both NADH and FA in the cytosol by lipid peroxidation leading both to NADH and FA oxidation followed by ATP synthesis in mitochondria).

1) Line 161 : Lehninger instead of Leninger

Ans) Thank you. It is fixed.

2) Line 279, Fig. 8. Perhaps MAS not in red, to be not confused with metabolites with major contribution to ATP production?

Ans) Thank you. It is changed to green in Fig 8.

3) Fig 8 right: I would make a combination of the two figures 8 with Fatty Acid -> FAO -> Acetyl-CoA (in TCA cycle)

Ans) Thank you for your comment. Arrow from FAO to TCA was inserted but not acetyl-coA due to the compact space in Figure 8.

4) I would keep the dotted blue line from Pyruvate to Acetyl-CoA and the blue line from (blue FAO) to NADH. A blue line from TCA cycle to NADH could be added.

Ans) Thank you. It is inserted in Figure 8.
